# Noninvasive Visualization of Tumor Blood Vessels within Hepatocellular Carcinoma by Application of Superb Microvascular Imaging to Contrast-Enhanced Ultrasonography

**DOI:** 10.3390/diagnostics14070678

**Published:** 2024-03-23

**Authors:** Yu Ota, Kazunobu Aso, Hideki Yokoo, Mikihiro Fujiya

**Affiliations:** 1Division of Gastroenterology, Department of Internal Medicine, Asahikawa Medical University, Asahikawa 078-8510, Japan; 2Division of Hepato-Biliary-Pancreatic and Transplant Surgery, Department of Surgery, Asahikawa Medical University, Asahikawa 078-8510, Japan

**Keywords:** hepatocellular carcinoma, lenvatinib, contrast-enhanced ultrasonography, sonazoid, superb microvascular imaging

## Abstract

The combination or sequential use of systemic therapies, such as lenvatinib and locoregional therapies, can improve the curability rate of hepatocellular carcinoma. This is based on the notion that lenvatinib remodels abnormal tumor vessels into normal vessels, potentially enhancing the efficacy of locoregional therapies. In this case report, we achieved noninvasive visualization of tumor blood vessels by applying superb microvascular imaging (SMI) to contrast-enhanced ultrasonography (CEUS). A man in his 80s with a borderline resectable hepatocellular carcinoma received preoperative therapy using lenvatinib. The patient achieved a complete response after lenvatinib therapy, underwent hepatectomy, and maintained a cancer-free status. CEUS and SMI revealed a decrease in tumor blood vessels at 1 week after lenvatinib administration and a decrease in tumor perfusion at 2 weeks. Although CEUS alone is adequate for noninvasive real-time evaluation of tumor perfusion, it is not sufficient to achieve accurate assessments of tumor blood vessels. We performed a noninvasive time-course evaluation of vascular normalization after lenvatinib administration by applying SMI. The evaluation of vascular normalization with lenvatinib therapy using CEUS and SMI can support the decision to proceed to conversion therapies.

Hepatocellular carcinoma (HCC) is the most common primary liver cancer. Advances in various systemic agents (e.g., lenvatinib) have substantially improved the treatment of HCC [1]. Although systemic treatments are typically restricted to patients with advanced- or intermediate-stage HCC who are not candidates for locoregional therapies, recent studies have focused on therapeutic strategies that solely target local control by adding resection [2], ablation [3], or transcatheter arterial chemoembolization (TACE) [4] to lenvatinib. These therapeutic strategies are based on the notion that lenvatinib remodels abnormal tumor vessels into normal vessels, a phenomenon referred to as “tumor vascular normalization”. The normalization of tumor vessels inhibits tumor growth and metastasis, while improving blood perfusion and oxygenation in tumor tissues; thus, it may enhance the efficacy of locoregional therapies [5].

A previous report from our hospital described conversion to surgery for recurrent HCC after lenvatinib treatment [6]. Additionally, a recent phase III randomized clinical trial of lenvatinib combined with TACE demonstrated long progression-free and overall survival [4]. However, long-term use of lenvatinib can cause adverse effects, lead to starvation of tumor vessels, and make TACE procedures difficult [7]. The optimal duration of lenvatinib administration before surgery or TACE has not been determined. Therefore, tumor vascular normalization with lenvatinib therapy should be accurately assessed to determine the appropriate timing for resection or TACE. Here, we describe a patient with HCC who achieved a complete response after lenvatinib therapy and was subsequently cured by hepatectomy. We evaluated the temporal changes in tumor blood vessels and arterial tumor perfusion induced by lenvatinib therapy using real-time contrast-enhanced ultrasonography (CEUS) and superb microvascular imaging (SMI).

A man in his 80s with steatotic liver disease and hypertension exhibited a liver lesion on ultrasonography in May 2023. He had no history of alcohol consumption, and he did not show markers for hepatitis B and C viruses. His serum alpha-fetoprotein (AFP) and Lens culinaris agglutinin-reactive AFP isoform 3 (AFP-L3) results were negative, but his protein induced by vitamin K absence or antagonist-II (PIVKA-II) level was elevated (70 mAU/mL). Multiphasic contrast-enhanced computed tomography (CECT) and magnetic resonance imaging (MRI) findings were suggestive of HCC (Figure 1 and Figure 2), and the chemical shift on MRI indicated the presence of intratumoral steatosis (Figure 2E,F; arrowhead). The tumor was confined to the liver, regional lymph nodes were normal, and there was no evidence of distant metastasis. However, the tumor was in contact with the middle hepatic vein (Figure 1C,D; arrow), and macrovascular invasion (Vv2) was suspected. Therefore, we initiated a multidisciplinary discussion with surgeons and physicians. According to the concept of resectability [8], the tumor was regarded as borderline resectable HCC. However, based on a previous report from our hospital [6], we speculated that the tumor would become resectable if a confirmed response to lenvatinib could be achieved for 8 weeks. Consequently, the patient received preoperative therapy rather than upfront surgical resection. Although he weighed 76 kg, lenvatinib was started at 8 mg/day (1 level of dose reduction) to maintain efficacy and limit toxicity.

We performed CEUS at baseline and at 1 and 2 weeks after initiation of lenvatinib administration. We assessed the lesion via B-mode scans using an Aplio i800 ultrasound system (Software Version 5.1, Canon Medical Systems, Otawara, Japan) with equipped with an i8CX1 transducer. Tumor size and arterial tumor perfusion were subsequently evaluated by CEUS with Sonazoid (GE Healthcare, Chicago, IL, USA); tumor blood vessels were assessed by SMI during the vascular phase of CEUS. A low-mechanical index (0.2–0.3) was set to avoid disrupting the microbubbles. Finally, three-dimensional ultrasonography volume data obtained by a manual sweep scan were imported to evaluate the response to lenvatinib therapy (Figure 3).

PIVKA-II level was slightly elevated (105 mAU/mL) at the 4-week assessment after initiation of lenvatinib administration, but Its level decreased (57 mAU/mL) at the 8-week assessment. The patient had achieved a complete response according to the modified Response Evaluation Criteria in Solid Tumors (mRECIST) (Figure 4) [9]. In previous report, surgery was aggressively performed when there was no tumor progression during the 8-week treatment period of lenvatinib for oncologically unresectable cases, which could achieve the high resection rate [10]. Thus, we speculated that the tumor would become resectable at a higher rate. Approximately 2 weeks after his last dose of lenvatinib, the patient underwent an extended medial segmentectomy (Figure 5). The patient did not develop discernable adverse events associated with lenvatinib therapy during the preoperative and perioperative periods. He rapidly recovered without complications, and was discharged 2 weeks postoperatively. AFP level was sustained within the normal range throughout the course. After the operation, PIVKA II levels decreased to within normal range; its level was 19 mAU/mL at 1 month and was 17 mAU/mL at 3 months, respectively. At the time of this writing, the patient was alive without disease 6 months after the operation; he had normal levels of AFP and PIVKA-II.

Systemic agents for HCC, including molecular targeted agents and immune checkpoint inhibitors, have been used alone and in combination in prospective phase III trials. Accordingly, many systemic agents have become available as treatment options for unresectable advanced-stage HCC and TACE-unsuitable intermediate-stage HCC [1,11,12]. Lenvatinib is an orally acting multikinase inhibitor that targets vascular endothelial growth factor receptors, fibroblast growth factor receptors, platelet-derived growth factor receptor-alpha, and RET and KIT proto-oncogenes. Lenvatinib demonstrated non-inferiority to sorafenib in terms of improving patient overall survival in the REFLECT trial [1]. The combination or sequential use of systemic therapies, such as lenvatinib and locoregional therapies, was recently reported to improve the curability rate in HCC [13].

We have presented a case of borderline resectable HCC in a patient who received preoperative therapy using lenvatinib rather than upfront surgical resection. The patient achieved a complete response after lenvatinib therapy; subsequent hepatectomy resulted in a cancer-free status. The tumor blood vessels and arterial tumor perfusion were evaluated by CEUS and SMI with Sonazoid to accurately assess the achievement of tumor vascular normalization with lenvatinib therapy. At the 1-week evaluation, CEUS and SMI revealed a decrease in tumor blood vessels without a decrease in arterial tumor perfusion. At the 2-week evaluation, these imaging examinations demonstrated a decrease in tumor blood vessels, along with a decrease in arterial tumor perfusion. Thus, lenvatinib induced tumor vessel normalization and subsequently improved tumor perfusion, as previously demonstrated in vivo [5]. To our knowledge, this is the first report to clinically and noninvasively evaluate the time course of vascular normalization after lenvatinib administration.

The concept of resectability is as follows [8]. First, unresectable is defined as the disease with distant metastasis or inability for macroscopic curative resection. Second, resectable is defined as a low-risk of post hepatectomy liver failure and the absence of macrovascular invasion. Finally, borderline resectable is defined as a high-risk of post hepatectomy liver failure and the presence of macrovascular invasion, such as Vp2-Vp4 and/or Vv2-Vv3. If CEUS and SMI demonstrate tumor vessel normalization with lenvatinib therapy, the therapy could subsequently achieve a complete response or partial response according to mRECIST. Once these responses are confirmed, we estimate that the tumor will become from borderline resectable to resectable at a higher rate.

The use of CEUS for evaluation of focal liver lesions is recommended by the American Association for the Study of Liver Diseases [14] and the European Association for the Study of the Liver [15]. We recently reported that CEUS is useful for diagnosing spontaneous necrosis in HCC, which is generally difficult to identify [16]. Additionally, we reported that CEUS can help to distinguish pseudoprogression from true progression of immunotherapy-treated HCC [17]. CEUS has become an important imaging modality in the diagnosis of liver tumors. Several previous studies evaluated changes in tumor vascularity with lenvatinib therapy by CEUS [18,19,20]. Kuroda et al. [18] observed the changes in tumor perfusion with time-intensity curve analysis before treatment and on day 7 by CEUS; they found that lenvatinib responders showed altered perfusion in the arterial phase. Eso et al. [19] also performed the quantitative assessment of tumor vascularity was performed with time-intensity curve. The quantitative tumor vascularity was defined as the area under the curve of the tumor area minus that of the background liver area, considering the effect of hemodynamic arrival changes. They reported their quantitative assessment of tumor vascularity was a useful predictor of therapeutic responses to lenvatinib. Kamachi et al. [20] evaluated changes in blood flow with the time-curve analysis before treatment and at 1 and 4 weeks after initiation of lenvatinib therapy. They also compared and evaluated the tumor blood flow and background liver blood flow. They reported that the therapeutic effect was significantly greater when the blood flow decreased by more than 50% after 1 week. These reports only described noninvasive real-time evaluations of arterial tumor perfusion rather than tumor vascular normalization. Although CEUS alone is adequate for evaluation of tumor perfusion, it is not sufficient for accurate assessment of tumor blood vessels. There have been no reports of tumor blood vessel visualization using CEUS, as performed in our case.

SMI is an image-processing technique developed by Canon Medical Systems [21]. This advanced ultrasound technology is based on an adaptive algorithm that can separate low-flow signals from overlaying tissue motion artifacts, thereby enabling visualization of microvascular flow [22]. Although CEUS can dynamically show real-time changes in arterial tumor perfusion and real-time changes in the local drainage area around the tumor, we also demonstrated the ability to visualize tumor blood vessels by applying SMI to CEUS. With respect to tumor blood vessel evaluation, Tachiiri et al. [7] observed tumor vessels on angiography; they confirmed decreases in vessel dilatation and tortuosity after lenvatinib therapy. Muraishi et al. [23] investigated tumor blood vessel shrinkage after lenvatinib therapy. They measured changes in the diameter of tumor blood vessels with sufficient size for evaluation by CECT. Progression-free survival was prolonged in patients who achieved tumor blood vessel shrinkage by lenvatinib [23]. However, angiography and CECT are invasive and difficult to use in patients with impaired renal function. CEUS is less invasive and can be used in patients with impaired renal function because Sonazoid is excreted by exhalation. Our CEUS and SMI findings are consistent with a previous angiography report, which revealed a decrease in tumor blood flow within the first 7 days of lenvatinib administration [7].

Matsuda et al. [24] observed that following long-term molecular targeted agent treatments such as lenvatinib, the diameters of hepatic arteries relatively decreased due to tumor ischemia and normalization of tumor blood vessels in the liver. In this study, the effect of TACE as a post molecular targeted agent treatment was limited and its effect on prognostic improvement was not able to be stated. Additionally, Yang et al. [25] evaluated the effect of different lenvatinib doses on promoting tissue perfusion and vascular normalization in both immunodeficient and immunocompetent mouse models. The underlying mechanisms were investigated by analyzing the vascular morphology of endothelial cells and pericytes. In this study, the adequate-dose lenvatinib effectively pruned the abnormal vessels and promoted the normalization of the remaining ones. However, the high-dose lenvatinib excessively pruned the functional vessels inside the tumor and resulted in the shortage of blood supply, which aggravated the hypoxia and perfusion. Based on these reports, the long-term or the high-dose lenvatinib therapy may not enhance drug delivery and the efficacy of TACE. Therefore, the decision to proceed to TACE should be made before lenvatinib therapy induces tumor ischemia and aggravates the hypoxia or perfusion. CEUS and SMI can be useful for the evaluation of vascular normalization prior to reduction in arterial tumor perfusion, and will help assist in determining the optimal timing of TACE.

In conclusion, this report described a patient with HCC, who achieved a complete response after lenvatinib therapy, and was subsequently cured by hepatectomy. We demonstrated the noninvasive visualization of tumor blood vessels by applying SMI to CEUS. The evaluation of vascular normalization with lenvatinib therapy using CEUS and SMI can support the decision to proceed to conversion therapies such as surgery or TACE.

## Figures and Tables

**Figure 1 diagnostics-14-00678-f001:**
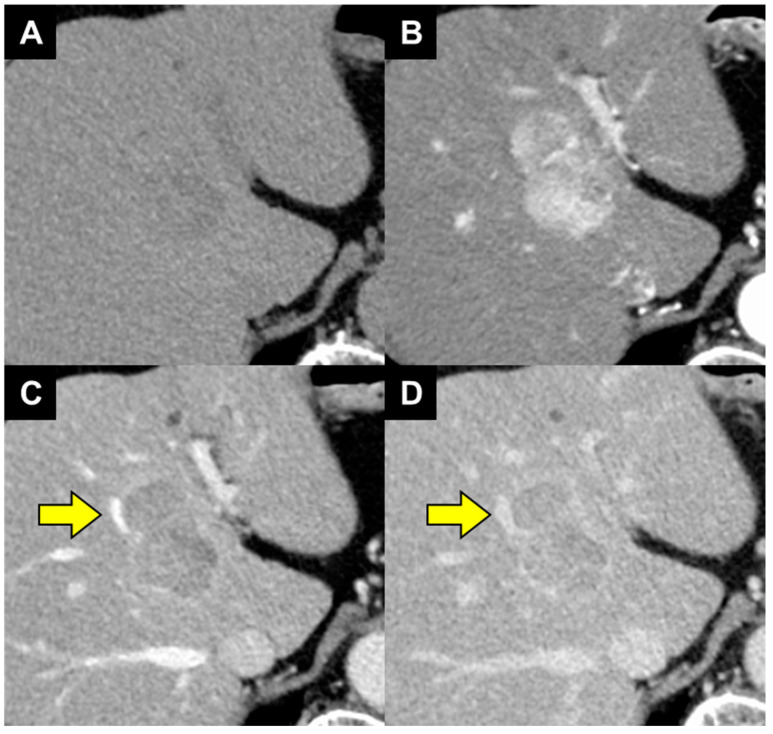
Abdominal contrast-enhanced computed tomography for initial diagnosis. (**A**) The plain scan revealed a round 45 mm diameter hypodense tumor in liver segment 4. (**B**) The tumor was enhanced in the arterial-dominant phase. The tumor was washed out in the (**C**) portal venous phase and (**D**) late venous phase. The tumor was in contact with the middle hepatic vein (arrow).

**Figure 2 diagnostics-14-00678-f002:**
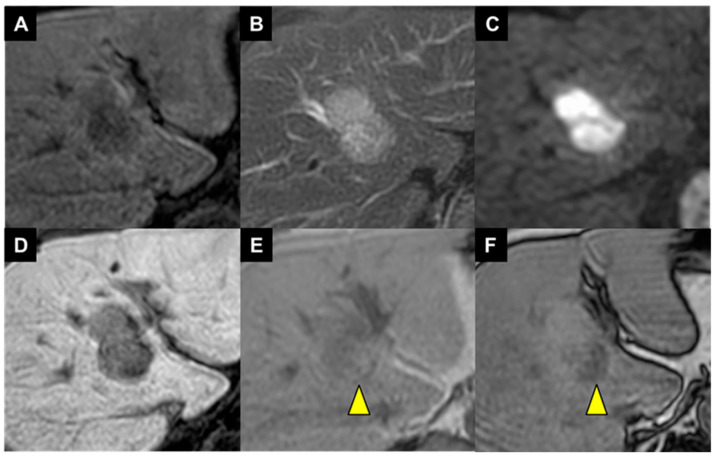
Gd-EOB-DTPA-enhanced magnetic resonance imaging. (**A**) The tumor was hypointense on plain T1-weighted imaging; it was hyperintense on (**B**) T2-weighted imaging and (**C**) diffusion-weighted imaging. The enhancement pattern of Gd-EOB-DTPA was identical to the pattern on CECT, and the tumor exhibited a defect during (**D**) the hepatobiliary phase. (**E**) In-phase T1-weighted imaging showed a partially hyperintense mass. (**F**) Opposed-phase T1-weighted imaging corresponding to the in-phase imaging revealed a decrease in tumor signal intensity ((**E**,**F**); arrowhead).

**Figure 3 diagnostics-14-00678-f003:**
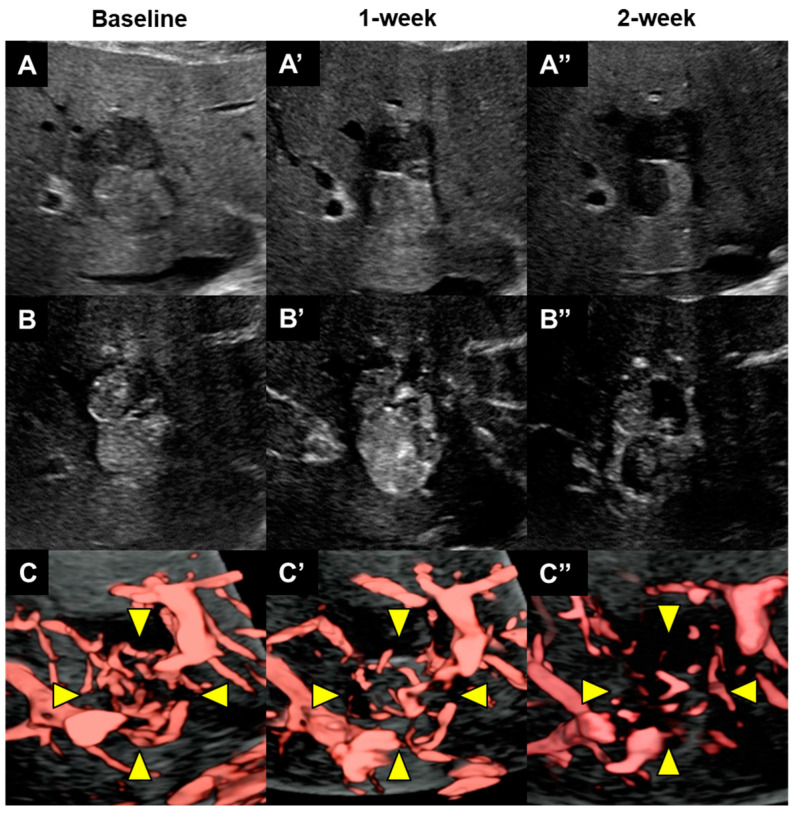
Changes in contrast-enhanced three-dimensional ultrasonography (CE3DUS) from baseline with Sonazoid. (**A**) B-mode, (**B**) vascular phase of CE3DUS, and (**C**) superb microvascular imaging (SMI) during the vascular phase. (**A**) Before lenvatinib administration, a B-mode scan showed the presence of S4 lesions that contained a mixed hypoechoic and hyperechoic area. The hyperechoic area appeared to reflect the presence of intratumoral steatosis, consistent with the chemical shift observed on MRI. CE3DUS showed (**B**) hyperenhancement in the tumor during the vascular phase and (**C**) blood vessels within the tumor exhibited a basket pattern during the vascular phase on SMI (arrowheads). At the 1-week assessment after initiation of lenvatinib administration, (**A’**) a B-mode scan showed no remarkable changes. CE3DUS revealed (**C’**) a decrease in tumor blood vessels on SMI (arrowheads) but (**B’**) no obvious changes in arterial tumor perfusion. At the 2-week assessment, (**A”**) a B-mode scan revealed newly detected hypoechoic change in hyperechoic areas. CE3DUS revealed (**C”**) further reduction in tumor blood vessels on SMI (arrowheads) and (**B”**) a decrease in arterial tumor perfusion during the vascular phase. The newly detected hypoechoic change and the decrease in arterial tumor perfusion seemed to reflect the tumor necrosis.

**Figure 4 diagnostics-14-00678-f004:**
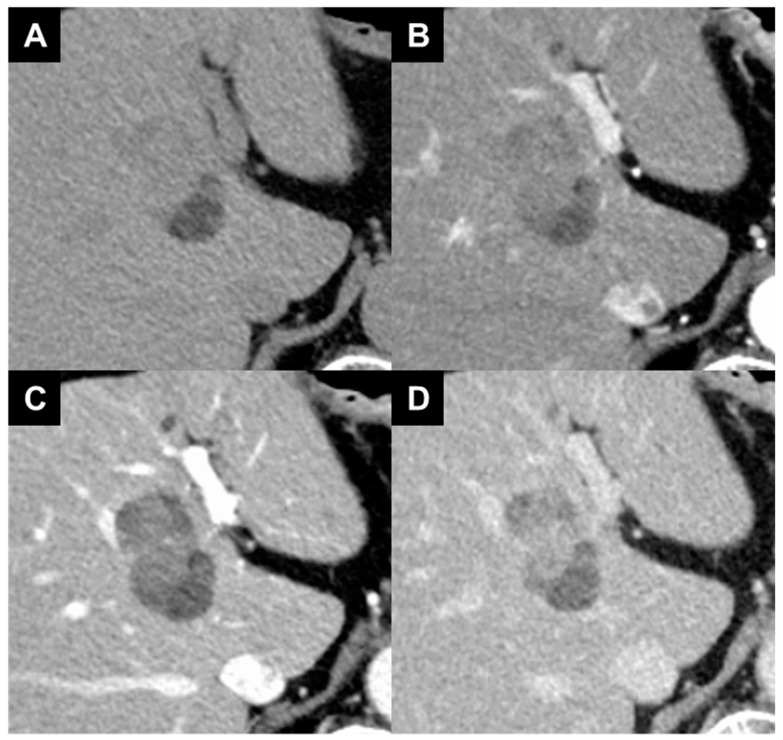
Abdominal contrast-enhanced computed tomography 8 weeks after initiation of lenvatinib administration. (**A**) Plain scan. (**B**) Arterial-dominant phase. (**C**) Portal venous phase. (**D**) Late venous phase. Tumor arterial enhancement was reduced and the tumor had not grown.

**Figure 5 diagnostics-14-00678-f005:**
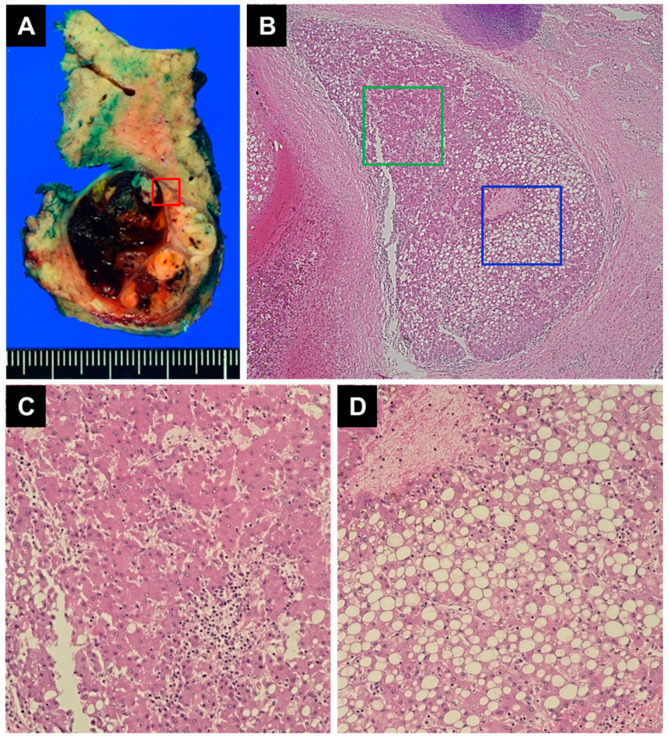
Macroscopic and histological images of the resected specimen. (**A**) Macroscopic findings of the resected specimen. The resected specimen contained a hepatic tumor measuring 3.5 cm. The cut surface of the tumor exhibited visible hemorrhage and necrosis. The red-bordered section is the fibrous capsule surrounding the nodule. (**B**) Microscopic findings of the red-bordered section in (**A**) (hematoxylin–eosin stain; magnification ×40). Viable cancer cells were present within the nodules. (**C**) Microscopic findings of the green-bordered section in (**B**) (hematoxylin–eosin stain; magnification ×100). The green-bordered section is the area of viable cancer cells. Microscopic analysis revealed well-differentiated hepato-cellular carcinoma without microscopic vascular invasion (vp0, vv0, va0, b0); it had negative surgical margins. (**D**) Microscopic findings of the blue-bordered section in (**B**) (hematoxylin–eosin stain; magnification ×100). The blue-bordered section is the area of intratumoral steatosis. Microscopic analysis revealed steatotic hepatocellular carcinoma with intratumoral steatosis. The non-tumor area exhibited features of steatohepatitis with bridging and pericellular fibrosis.

## Data Availability

The original data presented in the study are available on request from the corresponding author.

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
