# Peer review of "Noninvasive Visualization of Tumor Blood Vessels within Hepatocellular Carcinoma by Application of Superb Microvascular Imaging to Contrast-Enhanced Ultrasonography"

_diagnostics, 2024, doi:10.3390/diagnostics14070678_

Round 1

Reviewer 1 Report

Comments and Suggestions for Authors

This case report explores the potential synergistic effect of combining lenvatinib with locoregional therapies in the treatment of hepatocellular carcinoma, aiming to enhance the curability rate. The study introduces a novel approach by incorporating superb microvascular imaging (SMI) into contrast-enhanced ultrasound (CEUS) for noninvasive visualization of tumor blood vessels. The manuscript presents the case of an elderly patient with borderline resectable hepatocellular carcinoma who underwent preoperative therapy with lenvatinib, resulting in a complete response, hepatectomy, and sustained cancer-free status. The study holds promise, and with the suggested improvements, it has the potential to make a valuable contribution to the field.

  *** Detailed comments and questions ***

1- The effect of lenvatinib on the curability rate of hepatocellular carcinoma has already been investigated. What is the difference between this research and “Eso, Yuji, et al. A simplified method to quantitatively predict the effect of lenvatinib on hepatocellular carcinoma using contrast-enhanced ultrasound with perfluorobutane microbubbles Quantitative Imaging in Medicine and Surgery 11.6 (2021): 2766."? This paper should be mentioned in the references.

2- While the manuscript mentions the tumor being regarded as borderline resectable according to a specific concept of resectability, additional details on the criteria used for this classification would enhance clarity for readers.

3- Could the manuscript offer information on the specific criteria used to determine a "confirmed response" to lenvatinib, as it speculates that the tumor would become resectable if such a response is achieved for 8 weeks? Providing this information would strengthen the study's methodology.

4- There is no information about the ultrasound machine used. The specifications of the ultrasound machine used must be mentioned.

5- Clear definitions and criteria for tumor resectability and a "confirmed response" to lenvatinib should be provided to facilitate a better understanding of the study's methodology.

6- The manuscript briefly mentions previous reports, but a more thorough comparison with existing literature on lenvatinib therapy in HCC would strengthen the discussion and highlight the study's novelty.

Author Response

This case report explores the potential synergistic effect of combining lenvatinib with locoregional therapies in the treatment of hepatocellular carcinoma, aiming to enhance the curability rate. The study introduces a novel approach by incorporating superb microvascular imaging (SMI) into contrast-enhanced ultrasound (CEUS) for noninvasive visualization of tumor blood vessels. The manuscript presents the case of an elderly patient with borderline resectable hepatocellular carcinoma who underwent preoperative therapy with lenvatinib, resulting in a complete response, hepatectomy, and sustained cancer-free status. The study holds promise, and with the suggested improvements, it has the potential to make a valuable contribution to the field.

Response: Thank you for your time and consideration. We are very glad to hear kind comments from expert reviewer. We summarized a point-by-point response to the reviewer's comments along with an explanation. Responses were described by red characters. The changed and modified sentences in revised manuscript were also described by red characters.

Comments 1: The effect of lenvatinib on the curability rate of hepatocellular carcinoma has already been investigated. What is the difference between this research and “Eso, Yuji, et al. A simplified method to quantitatively predict the effect of lenvatinib on hepatocellular carcinoma using contrast-enhanced ultrasound with perfluorobutane microbubbles Quantitative Imaging in Medicine and Surgery 11.6 (2021): 2766."? This paper should be mentioned in the references.

Response 1: We appreciate the comments raised by the expert reviewer. Dr. Eso et al. performed the quantitative assessment of tumor vascularity was performed with time-intensity curve. Dr. Eso's previous report seems to reflect arterial tumor perfusion decrease rather than tumor vascular normalization. Their methods are similar to the reports of Dr. Kuroda et al. [PMID: 31148907] or Dr. Kamachi et al. [PMID: 34105904]. The difference between our report and Dr. Eso’s research is that we were able to visualize of tumor blood vessels within HCC. However, disadvantage of our manuscript is the lack of quantification. We have referred to the difference and have mentioned this paper in the References of the revised manuscript (page 6-7, line 178-183, 187-188).

Comments 2: While the manuscript mentions the tumor being regarded as borderline resectable according to a specific concept of resectability, additional details on the criteria used for this classification would enhance clarity for readers.

Response 2: We thank for the reviewer’s kind comment. In the classification of resectability for HCC, borderline resectable is first defined as a high-risk group of post hepatectomy liver failure assessed by indocyanine green clearance. Since macrovascular invasion is an indicator of oncologic disadvantage, resectable borderline is then defined as involvement of Vp2-Vp4 and/or Vv2-Vv3. As recommended, we have included a conceptual classification of resectability for HCC in the revised manuscript (page 6, line 159-164).

Comments 3: Could the manuscript offer information on the specific criteria used to determine a "confirmed response" to lenvatinib, as it speculates that the tumor would become resectable if such a response is achieved for 8 weeks? Providing this information would strengthen the study's methodology.

Response 3: We thank for the important points. According to LENS-HCC Trial, surgery was aggressively performed when there was no tumor progression during the 8-week treatment period of lenvatinib for oncologically unresectable cases, which could achieve the high resection rate. Thus, we speculate that the tumor would become resectable at a higher rate if complete response or partial response to lenvatinib treatment were achieved using mRECIST or RECIST version 1.1. As recommended, we described this point in the revised manuscript (page 4, line 107-111).

Comments 4: There is no information about the ultrasound machine used. The specifications of the ultrasound machine used must be mentioned.

Response 4: We incorporated the comments by the reviewer and have added information in the revised manuscript as below (page 3, line 80-88).

We performed CEUS at baseline and at 1 and 2 weeks after initiation of lenvatinib administration. We assessed the lesion via B-mode scans using an Aplio i800 ultrasound system (Canon Medical Systems, Otawara, Japan) with equipped with an i8CX1 transducer. Tumor size and arterial tumor perfusion were subsequently evaluated by CEUS with Sonazoid (GE Healthcare, Chicago, IL, USA); tumor blood vessels were assessed by SMI during the vascular phase of CEUS. A low-mechanical index (0.2–0.3) was set to avoid disrupting the microbubbles. Finally, three-dimensional ultrasonography volume data obtained by a manual sweep scan were imported to evaluate the response to lenvatinib therapy (Figure 3).

Comments 5: Clear definitions and criteria for tumor resectability and a "confirmed response" to lenvatinib should be provided to facilitate a better understanding of the study's methodology.

Response 5: We agree with the expert reviewer and have added information in the revised manuscript as below (page 6, line 159-167). Clear definitions and criteria for tumor resectability are as follows.

First, unresectable is defined as the disease with distant metastasis or inability for macroscopic curative resection. Second, resectable is defined as a low-risk of post hepatectomy liver failure and the absence of macrovascular invasion. Finally, borderline resectable is defined as a high-risk of post hepatectomy liver failure and the presence of macrovascular invasion. If CEUS and SMI demonstrate tumor vessel normalization with lenvatinib therapy, the therapy could subsequently achieve a complete response or partial response according to mRECIST. Once these responses are confirmed, we estimate that the tumor will become from borderline resectable to resectable at a higher rate.

Comments 6: The manuscript briefly mentions previous reports, but a more thorough comparison with existing literature on lenvatinib therapy in HCC would strengthen the discussion and highlight the study's novelty.

Response 6: We are thankful for your kind and suggestive comment. As the reviewer recommended above, we have first referred to the difference between our research and Dr. Eso's previous report and then have mentioned this paper in the References of the revised manuscript (page 6-7, line 178-183, 187-188).

In addition, we have added the following discussion (page 7, line 209-226).

Matsuda et al. observed that following long-term molecular targeted agent treatments such as lenvatinib, the diameters of hepatic arteries relatively decreased due to tumor ischemia and normalization of tumor blood vessels in the liver. In this study, the effect of TACE as a post molecular targeted agent treatment was limited and its effect on prognostic improvement was not able to be stated [PMID: 33622919]. Additionally, Yang et al. evaluated the effect of different lenvatinib doses on promoting tissue perfusion and vascular normalization in both immunodeficient and immunocompetent mouse models. The underlying mechanisms were investigated by analyzing the vascular morphology of endothelial cells and pericytes. In this study, the adequate-dose lenvatinib effectively pruned the abnormal vessels and promoted the normalization of the remaining. However, the high-dose lenvatinib excessively pruned the functional vessels inside the tumor and resulted in the shortage of blood supply, which aggravated the hypoxia and perfusion [PMID: 37545530]. Based on these reports, the long-term or the high-dose lenvatinib therapy may not enhance drug delivery and the efficacy of TACE. Therefore, the decision to proceed to TACE should be made before lenvatinib therapy induces tumor ischemia and aggravates the hypoxia or perfusion. CEUS and SMI can be useful for the evaluation of vascular normalization prior to reduction in arterial tumor perfusion, and will help assist in determining the optimal timing of TACE.

Reviewer 2 Report

Comments and Suggestions for Authors

Disadvantages of this manuscript

How did you use PIVKA-II? PIVKA-II is, however, a marker for monitoring recurrence

How was the patient's ultrasound? Describe liver lesion

What did liver markers look like?

More explanations about resectable, borderline tumors???

How many kilograms did the patient weigh? (for levantinib dose)

How do you explain that the AFP was negative?

Why was PET-CT not done?

The discussion and conclusions section needs to be included.

Author Response

Thank you for your time and consideration. We are very glad to hear kind comments from expert reviewer. We summarized a point-by-point response to the reviewer's comments along with an explanation. Responses were described by red characters. The changed and modified sentences in revised manuscript were also described by red characters.

Comments 1: How did you use PIVKA-II? PIVKA-II is, however, a marker for monitoring recurrence

Response 1: We thank for the reviewer’s kind comment. PIVKA-II in combination with AFP improves the detection of HCC, and PIVKA-II is valuable in the detection of HCC in AFP-negative HCC patients such as this case. As the reviewer suggested, PIVKA-II measurements are also useful for monitoring treatment outcomes and recurrence [PMID: 36710606].

We used PIVKA-II for the surveillance and treatment monitoring of HCC. In our case, pre-treatment PIVKA-II level was elevated, but its level decreased after treatment and remained normal level (page 4, line 104-106 and page 5, line 116-117).

Comments 2: How was the patient's ultrasound? Describe liver lesion

Response 2: We appreciate the comments raised by the expert reviewer. We have described the ultrasound findings of liver lesion (especially B-mode). The details have been stated in Figure3 Legends of the revised manuscript as below (page 4, line 90-103).

Changes in contrast-enhanced three-dimensional ultrasonography (CE3DUS) from baseline with Sonazoid. (A) B-mode, (B) vascular phase of CE3DUS, and (C) superb microvascular imaging (SMI) during the vascular phase. (A) Before lenvatinib administration, a B-mode scan showed the presence of S4 lesions that contained a mixed hypoechoic and hyperechoic area. The hyperechoic area appeared to reflect the presence of intratumoral steatosis, consistent with the chemical shift observed on MRI. CE3DUS showed (B) hyperenhancement in the tumor during the vascular phase and (C) blood vessels within the tumor exhibited a basket pattern during the vascular phase on SMI (arrowheads). At the 1-week assessment after initiation of lenvatinib administration, (A') a B-mode scan showed no remarkable changes. CE3DUS revealed (C') a decrease in tumor blood vessels on SMI (arrowheads) but (B') no obvious changes in arterial tumor perfusion. At the 2-week assessment, (A") a B-mode scan revealed newly detected hypoechoic change in hyperechoic areas. CE3DUS revealed (C") further reduction of tumor blood vessels on SMI (arrowheads) and (B") a decrease in arterial tumor perfusion during the vascular phase. The newly detected hypoechoic change and the decrease in arterial tumor perfusion seemed to reflect the tumor necrosis.

Comments 3: What did liver markers look like?

Response 3: We incorporated the comments by the reviewer and have included additional information in the revised manuscript as below (page 2, line 52-55 and page 4, line 104-106 and page 4-5, line 115-119).

AFP level was sustained within the normal range throughout the course. PIVKA-II level was slightly elevated (105 mAU/mL) at the 4-week assessment after initiation of lenvatinib administration, but Its level decreased (57 mAU/mL) at the 8-week assessment. After the operation, PIVKA II levels decreased to within normal range; its level was 19 mAU/mL at the 1-month and was 17 mAU/mL at the 3-month, respectively.

Comments 4: More explanations about resectable, borderline tumors???

Response 4: We thank you for your suggestive comments. We have added information in the revised manuscript as below (page 6, line 159-164).

First, unresectable is defined as the disease with distant metastasis or inability for macroscopic curative resection, Second, resectable is defined as a low-risk of post hepatectomy liver failure and the absence of macrovascular invasion. Finally, borderline resectable is defined as a high-risk of post hepatectomy liver failure and the presence of macrovascular invasion.

Comments 5: How many kilograms did the patient weigh? (for levantinib dose)

Response 5: Thanks for the suggestive comment. We have included additional information in the revised manuscript as below (page 2, line 66-67).

Although he weighed 76 kg weight, lenvatinib was started at 8 mg/day (1 level of dose reduction) to maintain efficacy and limit toxicity.

Comments 6: How do you explain that the AFP was negative?

Response 6: Thank you for the comments. Patients with NAFLD-HCC tend to have lower AFP levels than those with viral HCC [PMID: 36158261, PMID: 25148760]. We consider the possibility that this case was a NASH-related HCC and therefore AFP was negative.

Comments 7: Why was PET-CT not done?

Response 7: Thank you for the comments. At our hospital, when there are some findings that are suspicious of lymph node metastasis or distant metastasis on preoperative CECT, PET-CT is additionally performed to confirm the presence of accumulation in the lesions. In this case, we did not perform PET-CT because there was no lymph node swelling and no evidence of distant metastasis on CECT.

Comments 8: The discussion and conclusions section needs to be included.

Response 8: Thank you for the comments. Because this article type is "Interesting Images", we cannot include introduction, discussion, conclusions and so on.

Round 2

Reviewer 1 Report

Comments and Suggestions for Authors

The authors have addressed all my concerns, and questions, and the manuscript in this format is acceptable for publishing in the diagnostics journal.

Reviewer 2 Report

Comments and Suggestions for Authors

 Accept in present form